# The Role of Artificial Intelligence in Endoscopic Ultrasound for Pancreatic Disorders

**DOI:** 10.3390/diagnostics11010018

**Published:** 2020-12-24

**Authors:** Ryosuke Tonozuka, Shuntaro Mukai, Takao Itoi

**Affiliations:** Department of Gastroenterology and Hepatology, Tokyo Medical University, Tokyo 160-0023, Japan; tonozuka@tokyo-med.ac.jp (R.T.); s-mukai@tokyo-med.ac.jp (S.M.)

**Keywords:** artificial intelligence, deep learning, pancreas, computer-aided diagnosis, machine learning, endoscopic ultrasound, pancreatic cancer, convolutional neural network, deep neural network, support vector machine

## Abstract

The use of artificial intelligence (AI) in various medical imaging applications has expanded remarkably, and several reports have focused on endoscopic ultrasound (EUS) images of the pancreas. This review briefly summarizes each report in order to help endoscopists better understand and utilize the potential of this rapidly developing AI, after a description of the fundamentals of the AI involved, as is necessary for understanding each study. At first, conventional computer-aided diagnosis (CAD) was used, which extracts and selects features from imaging data using various methods and introduces them into machine learning algorithms as inputs. Deep learning-based CAD utilizing convolutional neural networks has been used; in these approaches, the images themselves are used as inputs, and more information can be analyzed in less time and with higher accuracy. In the field of EUS imaging, although AI is still in its infancy, further research and development of AI applications is expected to contribute to the role of optical biopsy as an alternative to EUS-guided tissue sampling while also improving diagnostic accuracy through double reading with humans and contributing to EUS education.

## 1. Introduction

Pancreatic cancer (PC) has the fifth-highest fatality rate among all carcinomas, with a five-year survival rate of approximately 6% [1]. Favorable long-term prognoses can be achieved through early detection and surgical resection, especially for tumors less than 1 cm in size, with a five-year survival rate of 80.4% [2]. Extracorporeal abdominal ultrasonography (US), computed tomography (CT), magnetic resonance imaging (MRI), endoscopic ultrasound (EUS), and endoscopic retrograde cholangiopancreatography (ERCP) have emerged as essential in the diagnosis of PC; technological advances have enabled them to provide precise imaging. Among these modalities, EUS enables observation of the pancreas with high spatial resolution, and the sensitivity of detection of PC using EUS has been reported to be 94% [3]. However, the diagnostic performance of EUS depends largely on the experience and technical abilities of the endoscopist. The American Society for Gastrointestinal Endoscopy recommends that EUS training should consist of at least two years of standard gastrointestinal (GI) fellowship (or equivalent training) and one year of pancreatic EUS training [4]. Screening by EUS is preferable in high-risk groups for PC, but increased numbers of tests may induce fatigue and inattention of endoscopists. However, because PC is rapidly progressive and fatal, there is concern that missing a diagnosis could have devastating consequences for patients. The implementation of artificial intelligence (AI) for image analysis has been studied in various disorders, some of which have already been used in clinical practice [5,6]. More advanced AI has been used in the field of gastrointestinal endoscopy, including for detection of colon polyps [7,8], discrimination between benign and malignant tumors [9], and evaluation of the depth of cancer invasion [10]. The application of AI into EUS for the pancreas is limited, and it is still in its developmental stage compared to other fields. Herein, we present a literature review with the aim of clarifying the progress and current prospects of EUS with AI for pancreatic disorders. In this review, for readers unfamiliar with AI, we first provide a brief overview and refer readers to cited works for precise details and further information.

## 2. Overview of Artificial Intelligence in Diagnostic Imaging

### 2.1. From Artificial Intelligence to Deep Learning

There are many definitions of AI, but the concept can be simply described as computer programs developed by humans and equipped with analogs of the thoughts, judgments, and reactions that take place in the human brain. Under this definition, one way to create AI is machine learning (ML), which refers to a method of learning that utilizes a large amount of input data and finds the various and complicated patterns or features that occur within it. There are three types of ML: supervised learning, where the program learns by correcting for differences between correct data and program output corresponding to the input data; unsupervised learning, where the program learns without correct data, assuming stationarity of the input, according to its distribution and similarity; and reinforcement learning, where the program learns through adjustments, by not giving the program direct correct data, but instead by evaluating and rewarding the output. Supervised learning has mainly been used in diagnostic imaging, typical examples being artificial neural network (ANN) [11], naïve Bayes [12], logistic regression [13], decision tree [14], random forest [15], and support vector machine [16] (SVM, which is described in detail in Section 2.3.). In early ML development, human designers struggled to create features such as shape and density information from images in ingenious ways. However, other ML techniques allow such features to be created by themselves through a learning process, which can save considerable time and effort. Deep learning (DL) is one type of ML that has been developed to further this goal, and it involves the development of ANNs that realize AI through the use of multi-layered and complex structures (Figure 1). ANNs are based on the perceptron, which was first published in 1958 as an attempt to mimic the human brain’s neural circuits (Figure 2a) [11]. ANNs apply this perceptron to represent the data received from the input layer as it passes through the hidden (i.e., middle) layer and finally the output layer to represent the desired output, and the neurons (i.e., nodes) in each layer are connected by a weight coefficient that indicates the strength of the connection, bias, and an activation function such as a sigmoid (logistic) function or hyperbolic tangent function (tanh) (Figure 2b) This flow of information from input to output is called forward propagation. Through a proper training (learning) process, the network can adjust the value of the weights of the connections to obtain the best results. Then, based on the errors (loss) between the output and the correct data, the slope of the loss function is calculated and then propagated through each layer toward the input layer. In each layer, the weight coefficients and biases are adjusted based on the slope of the loss function. This is referred to as the backpropagation algorithm [17].

DL is a technology that utilizes a deep neural network (DNN), which is an ANN with four or more layers obtained by increasing the number of hidden layers. This enables handling problems that are not linearly separable and cannot be solved by the simple perceptron, as well as complex tasks and large amounts of data. (Figure 2c). However, we should be aware of issues such as overfitting and the vanishing gradient problem, which can occur as the DNN becomes deeper. Overfitting is a general phenomenon in ML where the model fits too well to the training samples, resulting in a low accuracy rate when evaluating unknown samples; in other words, the model is optimized for training data only and has no generality. To prevent overfitting, various efforts have been made to increase the amount of training data and regularize and simplify the models in ML model creation. The vanishing gradient problem is a phenomenon of multi-layered ANNs, in which the gradient approaches zero as it nears the input layer and finally disappears, resulting in a loss of learning. One way to solve this problem is to use the rectified linear unit (ReLU) instead of the sigmoid function for the activation function, which has been used in many DL studies. The ReLU function output is 0 if the input is a negative value, and the output is *x* if the input *x* is a positive value, such that the slope is 1, thus avoiding the vanishing gradient problem. 

In conjunction with the significant advances in computer technology, such as graphics processing units (GPUs) with high computational power, the acquisition of large amounts of data through the development of the internet, and the development of various DL algorithms, including convolutional neural networks (CNNs) [18], autoencoders [19], and generative adversarial networks [20], have flourished, and various algorithms have emerged to further improve accuracy, solve enormous computational problems, and increase flexibility in learning. In particular, CNNs have been confirmed to be far superior to conventional image recognition methods and are now commonly used in medical imaging as well [21]; additional details about CNNs are described in 2.4 below.

### 2.2. Computer-Aided Diagnosis

Diagnosis based on image processing by computers is referred to as computer-aided diagnosis (CAD); the use of DL has become a mainstream AI application [22]. There are various roles in diagnostic imaging using computer systems, primarily computer-assisted detection (CADe) for lesion detection within an image, computer-assisted diagnosis (CADx) for differentiation (classification) of lesions, and segmentation for extraction of the area, including the contour of the object, which facilitates identifying the detailed delineation of the lesion and the category (i.e., lesion or organ) to which individual pixels belong. The basic technologies involved in the CAD schemes are (i) image processing such as normalization, (ii) image input, (iii) feature extraction, and (iv) results for detection or classification. In conventional CAD, each step of the process is conducted by human researchers themselves or with the help of computers, while DL-based CAD can automate this sequence of steps, end to end, through a learning process [23]. There are first reader, second reader, and concurrent reader forms of computer support, whereas the second reader type is the one most commonly found in CAD today. This is diagnosed first by the doctor, as usual; then, the computer results are reviewed, providing the possibility to change the interpretation and diagnose as necessary. On the other hand, a concurrent reader-type CAD system that refers to the CAD output at the same time as the diagnosis and a CAD system similar to the first reader type that makes a decision before the specialist’s diagnosis have also been developed [5].

### 2.3. Support Vector Machine 

SVM is one of the supervised learning ML algorithms [16] (Figure 3). The basic concept is to classify data belonging to two categories by creating a boundary. When there are two types of data for classification, the boundary is a line; when there are three types, the boundary is a plane; and when there are four or more types, the boundary is a hyperplane boundary, which is collectively referred to as a “separating hyperplane”. A support vector represents the data closest to the separating hyperplane, and the distance between the support vector and the separating hyperplane is referred to as a margin; an SVM achieves classification by calculating the separating hyperplane such that this margin is maximized. (Figure 3a). If the data are linearly separable, this can be accomplished by a very simple calculation; however, in practice, including in imaging, data are generally not linearly separable. In such cases, soft-margin SVM and the kernel method are used to deal with the data. Essentially, soft-margin SVM allows a few anomalous expression profiles to fall on the “wrong side” of the separating hyperplane. Hence, introducing the soft margin necessitates introducing a user-specified parameter that roughly controls how far across the boundary data are allowed to be. There is a trade-off between hyperplane violations and margin distance, therefore ML classification is accomplished by trying to maximize the margin while allowing hyperplane violations as much as possible (Figure 3b). On the other hand, the concept of the kernel method is to map features to a high-dimensional feature space that is linearly separable (Figure 3c). However, this requires a huge amount of computation because features must be mapped to a number of dimensions as large as the number of data, so replacing the inner product of the non-linear map with a kernel function reduces the computational complexity significantly. This technique is called the kernel trick [24], for which several kernel functions can be used, such as Gaussian kernels, polynomial kernels, and radial basis function kernels. There are four main advantages of SVM: (i) the kernel trick allows application to nonlinear problems, (ii) the solution is theoretically unique due to the explicit criterion of margin maximization, (iii) generalizability is high due to margin maximization, and (iv) fewer parameters must be set in advance compared with neural networks. Nevertheless, SVM has several key disadvantages: (i) the effective features for classification must be determined manually, (ii) the learning time is huge when the number of data is large, (iii) some ingenuity is required to perform multi-class classification because it is basically a two-class classification method, and (iv) the output data do not give the probability of belonging to a classification category.

### 2.4. Convolutional Neural Network

A CNN is a DL algorithm that was developed based on the processing of human visual signals (specifically, simulations of the principles of the cortical visual cortex V1 surrounding the avian sulcus in the occipital lobe). CNNs have made significant contributions to computer vision (the process by which a computer obtains and recognizes information from a series of images or videos) and continue to play a pivotal role in medical imaging analysis. In contrast to the feature extraction algorithms used in traditional CAD, which require human trial and error, CNNs use the image itself as an input and automatically learn to identify the most suitable features. In addition, conventional ANN has the problems of overfitting and vanishing gradient as the number of hidden layers increases; however, with CNNs, the layers are not fully connected, and the weights are adjusted for multiple data because multiple pixels in the input image (input feature map) share a single weight, thus preventing overfitting. The typical CNN consists of the following: (i) a convolutional and pooling layer for extracting distinctive features, and (ii) a fully connected layer for the overall classification. The input images are filtered by a number of specific filters automatically to extract the distinctive features to create multiple feature maps (Figure 4a). This operation for filtering is called convolution, and the process of training the convolutional filters to create the best feature maps is essential for success with CNNs. These feature maps are compressed to a smaller size in the pooling layer, and these convolution and pooling layers are repeated many times. Finally, a fully connected layer combines all the features to obtain the final result (Figure 4b). ReLU (used as an activation function to avoid the vanishing gradient problem) and data augmentation (used to increase the number of training images by performing micro-geometric transformations such as image inversion, resizing, and shifting) are commonly used to improve the generalizability of CNNs. In addition, the dropout technique is often used to prevent overfitting by randomly disabling some units in a layer and performing backpropagation in the remaining units [25]. Various CNN architectures, such as LeNet [26], Alexnet [27], GoogLeNet [28], VGGNet [29], and ResNet [30] have been proposed, and applications of them have been used in many kinds of research. 

### 2.5. Validating Methods in Machine Learning

An important aspect of ML, including DL, is that the model learns the data such that it can accurately make predictions and classifications using unknown data. In other words, the model needs to acquire generalization performance. For this purpose, in the development of ML, the generalization performance of the model is necessarily validated, and there are several ways to achieve this, including the following approaches.

#### 2.5.1. Hold out Validation

Some cases are randomly selected from the initial samples to form test cases, while the remaining cases are used for training. In general, it is often one-half, one-quarter, or one-nineth of the initial samples that are used for test cases. However, this is not suitable for the validation of a study with small amounts of data because of data bias.

#### 2.5.2. K-Fold Cross-Validation

In K-fold cross-validation, the samples are divided into K groups, one of which is used as a test, and the remaining K-1 group is used for training. In cross-validation, each of the K groups of samples is tested k times, and the average of the tests is the cross-validation result. Although this method, depending on the number of patterns, is more computationally intensive than hold-out validation, it is currently widely used owing to its confidence and efficiency. 

#### 2.5.3. Leave-One-Out Cross-Validation 

Only one sample is extracted from each of the N samples and used as test data, and the model is trained on the remaining N-1 samples and verified N times. More training data can be obtained than in the above two methods, but because the amount of computation is enormous in proportion to the number of samples, it is limited to studies with a relatively small number of samples.

## 3. Literature Search

Two authors (R.T. and S.M.) used the Google search engine, Google Scholar, MEDLINE via PubMed, OVID, Cochrane, and Scopus to search for articles published up to September 2020 using the following keywords alone or in combination: endoscopic ultrasound, endosonography, artificial intelligence, deep learning, computer-aided diagnosis, machine learning, and pancreas. We carefully selected articles from those searches that accurately and clearly described the application of CAD to EUS for the pancreas. We also manually retrieved eligible studies from the references in the review articles. All articles included were in written in English only.

## 4. Computer-Aided Diagnosis for Pancreatic Endoscopic Ultrasound

A summary of the studies that have evaluated the use of CAD in diagnostic EUS is presented in Table 1. The first report of using CAD for EUS in the pancreas was in 2001 by Norton et al. [31]. In the following 17 years, there were some reports of conventional CAD, in which researchers made computer-based extraction and selection of appropriate features, and then analyzed them with an ML algorithm. Finally, in 2019, DL-based CAD was introduced in this area.

### 4.1. Conventional Computer-Aided Diagnosis 

Norton et al. [31] reported the first use of CAD based on digital image analysis (DIA) of EUS images in 2001. This study included a total of 35 patients with a histological diagnosis, including 14 with focal chronic pancreatitis (CP) and 21 with PC, who were randomly selected from patients examined using radial EUS. Representative images (region of interest, ROI) data for each case were then extracted by humans and inputted into a computer program. This computer program scanned rows of grayscale pixels from the digitized and described image characteristics such as degree of variation in grayscale between adjacent pixels, grayscale variation over a length (“run”) of pixels, and overall brightness. Then, the four features associated with “run” were judged to support maximizing the distinction between PC and CP. After each lesion was represented on a two-dimensional grid according to the results, the authors attempted to establish an arbitrary division that would allow for a strong distinction between the two types of disease. Finally, Norton et al. were able to maximize their overall diagnostic sensitivity to 89%. When the sensitivity for malignant diseases was set to 100%, aiming to reduce the number of missed malignancies, the overall diagnostic accuracy was 80%. This result was remarkably similar to the 85% accuracy obtained by the actual EUS test and the 83% accuracy of the diagnosis by other observers who watched the video of the test (blinded assessment). In summary, this study by Norton et al. [31] cannot be described as AI-CAD for contemporary clinical applications because the computer algorithm was very simple, the number of data points was very small, and the EUS images were of very low resolution. However, this study showed that the analysis of EUS images using DIA was feasible and comparable to human interpretation in different pathological situations, establishing a foundation for later AI research in EUS imaging.

Das et al. [32] studied the diagnosis of PC by ANN with the input of parameters acquired by DIA of EUS images in 2008. They selected the ROI of each disease from EUS images from a total of 56 patients including 22 patients with a normal pancreas (NP), 12 with a CP, and 22 with PC. Then, from the ROI image data of 110 NP, 99 CP, and 110 PC images, texture analysis was performed, which is one of the techniques used to evaluate the distribution and spatial variation of pixel intensities. Eleven independent features were extracted from the 228 extracted texture parameters using principal component analysis based on the correlations between each parameter (cumulatively explaining 96% of the variation in the original 228 parameters). They built a CAD model for the diagnosis of PC using a multilayered perceptron neural network that consisted of a single input layer, including nine inputs based on the eleven parameters, and nine hidden layers. The single output layer was a dichotomy of PC status (presence/absence). Of all the data, a randomly selected 50% was used for training, and the remaining 50% was used for validation. Validation results showed a sensitivity of 93%, specificity of 92%, positive predictive value (PPV) of 87%, and negative predictive value (NPV) of 96%. The area under the receiver operating characteristic curve (AUROC) was also high, at 0.93. Additionally, a model of differentiation between NP and CP individuals was developed and validated, achieving sensitivity and specificity of 100%. In summary, Das et al. [32] reported the first ML results for EUS images of the pancreas using a multi-layered neural network, suggesting that this is possible, although challenges presented by a lack of pathological evidence in CP and NP individuals and small sample sizes remained. 

Zhang et al. [33] in 2010 performed various digital processing procedures with EUS images to select better texture features and then used them to build an SVM prediction model for differentiating PC from non-PC cases. A total of 216 patients, including 153 PC and 63 non-PC patients (20 NP and 43 CP patients), were confirmed with a final diagnosis by definitive cytology, surgical pathology, and a clinical follow-up more than 12 months later. They set the ROI from each EUS image and extracted texture features from the ROI. The authors used multifractal dimensional features, a quantitative measure of fractality (self-similarity) and complexity, as texture features. The better combinations of features were then examined using a sequential forward selection process and a Bayesian classifier. In addition, they combined 20 other frequently used texture features with multifractal features to investigate a better combination of features. As a result, they found that they obtained the highest classification accuracy with seven-dimensional multifractal features. They also found that by adding nine other texture features to this, the classification accuracy reached 99.07%. These were then introduced into the SVM, and 50 random experiments were conducted, and the accuracy, sensitivity, specificity, PPV, and NPV were each high, at 97.98%, 94.32%, 99.45%, 98.65%, and 97.77%, respectively. This report was the first to suggest the possibility of using CAD with SVM introducing multiple features based on DIA. In summary, this study by Zhang et al. [33] suggests that SVM is a useful method for diagnosing PC with EUS images and could be implemented for rapid non-invasive screening of pancreatic disease. 

Săftoiu et al. (2012) [34] reported a prospective, blinded, multicentric study of the accuracy of real-time EUS elastography in focal pancreatic lesions. A total of 258 patients, 47 with CP and 211 with PC from 12 European hospitals, with final diagnoses obtained by cytology, histology by EUS-guided fine needle aspiration (EUS-FNA), and/or surgical pathology and/or a clinical follow-up after a minimum period of six months, were enrolled in the study. Three 10 s videos per patient were obtained, and ROI was set to include 50% of the surrounding structures. The image analyst used computer software to cut 125 still images displaying 256 colors from one dynamic elastography movie to obtain 125 hue histograms. The ANN used was the multilayer perceptron with two hidden layers trained using a back-propagation algorithm. During training, the number of neurons in each hidden layer was varied from 1 to 128, with the final number of neurons being 55 in the first hidden layer and 34 in the second layer, which gave the best accuracy. They ran the algorithm 100 times in a complete cross-validation cycle (ten-fold cross-validation) to avoid the inherent bias induced by its heuristic nature, and sensitivity, specificity, PPV, and NPV values were 87.59%, 82.94%, 96.25%, and 57.22%, respectively. The corresponding average area under the receiver operating characteristic curve (AUC), over the 100 different computer runs of a complete cross-validation cycle, was 0.94. In summary, Săftoiu et al. [34] successfully used ML with ANN to objectively interpret EUS elastography, a traditionally subjective method of testing. 

In 2013, Zhu et al. [35] (from the same group as Zhang et al. [33]) developed another texture analysis-based CADx model using a SVM classifier to differentiate PC from CP cases. A total of 388 patients, including 262 PC and 126 CP patients, were randomly extracted from the single-center EUS-FNA database. From these EUS images, the ROI was set for each of them, and 105 parameters in nine categories were extracted using image analysis software as histograms. Among them, 25 better feature combinations were selected by the distance between class algorithm, which is a method that determines that the greater the distance between classes (meaning the difference between the medians of the features of the two classes), the greater the accuracy of the classification performance. In this study, a total of 388 patients were classified into a training set and a test set according to two different methodologies: the half-and-half method of hold-out validation and the leave-one-out method. Then, feature selection was performed by the sequential forward selection (SFS) algorithm to reduce the training time, improve the interpretability and accuracy of the model, and prevent overfitting. The SFS algorithm is a method used to select better combinations of features that selects the combination with the best discrimination performance after adding features one by one. Using this method, Zhu et al. were able to achieve optimal classification accuracy, with a classification error rate of 4.38% for 16 combinations out of 25 features. These features were then introduced into the SVM and examined using two different training/testing classification methods. In the half-and-half method, the accuracy, sensitivity, specificity, PPV, and NPV were 93.86%, 92.52%, 93.03%, 91.75%, and 94.39%, respectively, while in the leave-one-out method, they were 94.16%, 91.55%, 95.07%, 93.67%, and 96.98%, respectively. In summary, Zhu et al. [35] demonstrated the potential of ML with SVM by carefully identifying combinations with high classification performance from a number of texture features.

In 2015, Săftoiu et al. [36] generated time-intensity curves (TICs) based on data derived from dynamic contrast-enhanced EUS performed on solid pancreatic masses and analyzed them using multilayered ANN to classify CP from PC cases. The study prospectively enrolled consecutive patients in five European hospitals (Romania, Denmark, two German centers, and Spain) with solid pancreatic masses diagnosed cytologically or pathologically, and ultimately enrolled a total of 167 patients, including 112 PC patients and 55 CP patients, for the analysis. A randomly selected 70% of cases were used for the ANN training set, 15% for the validation set, and 15% for the test set. The validation set in this study was used to measure the generalization of this ANN and was responsible for stopping the training when it was confirmed that sufficient generalization had been reached. From the video data of each case, two ROIs were created per case. One was set to include 50% of the lesion and the other to include the surrounding normal pancreatic parenchyma. Seven parameters from TIC curve analysis were then quantified by the software: peak enhancement, the wash-in area under the curve, rise time, mean transit time, time to peak, wash-in rate, and wash-in perfusion index. Among seven parameters, the values of peak enhancement (*p* = 0.0001), wash-in area under the curve (*p* = 0.0009), wash-in rate (*p* = 0.0008), and wash-in perfusion index (*p* = 0.0027) were determined to be significantly different between CP and PC cases. Then, they constructed a feed-forward network consisting of an input layer with seven neurons, a single hidden layer with a sigmoid function, a backpropagation algorithm, and an output layer with two neurons. In terms of the diagnostic ability of ANN introduced with all seven parameters, the sensitivity, specificity, PPV, and NPV were 94.64%, 94.44%, 97.24%, and 89.47%, respectively. In summary, Săftoiu et al. [36] suggest that CAD by ANN using the parameters obtained from their TIC analysis may provide additional diagnostic value to human qualitative CE-EUS interpretation and EUS-FNA results.

Ozkan M et al. [37] focused on age-dependent pancreatic changes and proposed a new CAD system using ANN to distinguish between PC and NP cases in three age groups. The classifier in the designed system can receive EUS images for all age groups together as input for training and testing, as well as receive them separately. From the EUS images of 172 people who underwent EUS, 202 images of PC cases and non-PC cases for which pathological diagnosis was obtained by EUS-FNA 130 images were used in this study; 11 patients with PC (21 images) and 29 non-PC patients (47 images) were included in the <40-year-old group, and 36 patients with PC (41 images) and 22 non-PC patients (34 images) were included in the 40-to-60-year-old group, while the >60-year-old group included 46 patients with PC (140 images) and 28 non-PC patients (49 images). Firstly, in the image preprocessing phase, multiple image processing filters were used to improve image quality, remove noise, and remove characteristic edges in the image. Then, after image reduction, two gastrointestinal endoscopists identified and segmented the lesions in the segmentation phase. Next, in the feature extraction phase, 122 digital features were extracted from six categories. However, introducing all 122 features into the ANN would have been time-consuming, so the researchers used the Relief-F feature reduction method to reduce the number of features to 20 for optimal results. The ANN model in this study was a multilayered feed-forward perceptron with three layers: the input layer, a single hidden layer, and an output layer. Additionally, the ANN was trained by feeding digital features of cancer and non-cancer separately. In the under-40 group, 13 cancer and 30 non-cancer images were used for training, while the remaining images (8 cancer and 17 non-cancer) were used in the test data, and the performance with this age group had an accuracy of 92%, sensitivity of 87.5%, and specificity of 94.1%. In the 40-to-60-year-old group, 27 cancer and 22 non-cancer images were used for training, and the remaining images (14 cancer and 12 non-cancer) were used in the test data; the performance for this age group reached 88.5% accuracy, 85.7% sensitivity, and a specificity of 91.7%. In the >60-year-old group, 110 cancer images and 31 non-cancer images were used for training, and the remaining images (30 cancer and 18 non-cancer) were used in the test data; the performance with this age group reached 91.7% accuracy, 93.3% sensitivity, and a specificity of 88.9%. On the other hand, for the entire patient population, 160 cancer, and 100 non-cancer images were used for training, and the remaining images (42 cancer and 30 non-cancer images) were used in the test data. The overall performance for the combined population achieved 87.5% accuracy, 83.3% sensitivity, and 93.3% specificity. They concluded that the performance would be better if the analysis was performed by age group.

### 4.2. Deep Learning-Based Computer-Aided Diagnosis

Kuwahara et al. [38] in 2019 reported a predictive model for malignant intraductal papillary mucinous neoplasm (IPMN) in EUS images using CNNs, a DL method. A total of 50 IPMN patients who underwent surgical resection were included in the study, with low- and intermediate-grade dysplasia defined as benign, and high-grade dysplasia and invasive carcinoma defined as malignant IPMN. A total of 3970 still images from 50 patients were collected and converted into levels on a grayscale (0–255) for each pixel, which was increased to 10,000 by data augmentation and then fed into the CNN model. The original CNN algorithm was based on the ResNet50 algorithm with max-pooling and global average pooling layers. To speed up the training and prevent overfitting, various methods, such as batch normalization, stochastic depth, early stopping, data augmentation, random cropping, and random erasing were used. The optimization algorithm used to train the network weights was a momentum stochastic gradient descent estimation implementation. They set the output value to be a continuous variable from 0 to 1, with a value close to 1 indicating a high probability of malignancy, which they called the AI value. In this study, ten-fold cross-validation (with a training/test set ratio of 90/10, conducted 10 times) was used to verify the validity of this algorithm. The area under the ROC curve for the ability to diagnose malignancies of IPMNs via AI malignant probability was 0.98 (*p* < 0.001). When an AI malignant probability of 0.41 was used as a cutoff point based on the ROC analysis, the sensitivity, specificity, and accuracy of the AI malignant probability were 95.7%, 92.6%, and 94.0%, respectively; its accuracy was higher than human preoperative diagnosis (56.0%), as well as that based on the presence of mural nodules ≥5 mm (68.0%). Multivariate analysis including patient backgrounds and imaging findings, and human preoperative diagnosis, and this AI value showed AI malignant probability to be the only independent factor for IPMN-associated malignancy (odds ratio, 295.16; 95% confidence interval, 14.13–6165.75; *p* < 0.001). Thus, Kuwahara et al. [38] concluded that the use of AI is recommended for objectively assessing the preoperative malignancy of IPMNs.

Zhang et al. [39] built a DL-based image classification model using multiple deep CNNs (DCNNs) to utilize the “station approach” in pancreatic EUS, which had been established as the standard scanning procedure to allow endosonographers to recognize landmark images and, underneath them, to perform a comprehensive examination of the pancreas and biliary tract. In this study, the station approach of EUS in the pancreas was divided into six stations: (1) abdominal aorta; (2) the pancreatic body; (3) pancreatic tail; (4) confluence; (5) pancreatic head from the stomach; and (6) pancreatic head from the descending part of the duodenum. This DCNN base model, named BP MASTER (pancreaticobiliary master) system, consisted of four DCNNs. DCNN1 was trained to pick up only EUS images from among the white light images and EUS images in the video. DCNN2 was trained to filter out unqualified EUS images, which were defined as images that could not be classified into a pancreas station. Thus, for DCNN1 and DCNN2, 19,486 standard station and 15,684 unqualified EUS images were used for training, and 3897 standard station and 3137 unqualified EUS images were used for testing. DCNN3 classified qualified EUS images into the six stations. DCNN4 segmented landmarks, such as the pancreas and blood vessels, in the six stations. For the training of DCNN3 and DCNN4, 247 examinations (19,487 images) were used, while 44 examinations (1920 images) and 29 examinations (180 images) were used for internal testing and as a comparison data set with endoscopist diagnoses, respectively. The authors also performed 28 video validations that were prospectively registered separately at the same hospital, and external validations based on 109 examinations (768 images) that were performed at other hospitals were also considered. They then conducted the following three assessments: (i) examination of the performance of each DCNN; (ii) comparative testing of three endoscopy experts and models; (iii) a crossover study to test the system effect on reducing the difficulty in ultrasonographics interpretation among trainees. Eight trainees were randomly divided into group A, which first evaluated the videos and images without BP MASTER augmentation, and group B, which first evaluated the videos and images with BP MASTER augmentation. After a washout period of two weeks, the arrangement was reversed. The average accuracy of the six stations determined by DCNN3 was 0.942, 0.824, and 0.862 for the internal and external test data sets and the video test set, respectively. For the evaluation of the segmentation performed by DCNN4, the Dice coefficient, which indicates the degree of agreement for the set, was 0.715, and recall (true positives / [true positives + false negatives]) at 50% intersection over union (50% IoU, the overlap between the predicted and ground-truth bounding box is more than 50%) and precision (true positives / [true positive + false positives]) were 100% and 52.2%, respectively. When this model was applied to 20 patients (180 photographs), the accuracy and the Dice coefficients for the pancreas and blood vessels were 90%, 0.77, and 0.813, respectively, comparable to those of three endoscopists who also evaluated this group. The model also achieved substantial interobserver agreement with the three endoscopists (κ = 0.846, 0.853, and 0.826, respectively). There was a statistically significant increase in the mean accuracy (8.4%) and the mean Dice coefficients of both blood vessels (8.6%) and pancreas segmentation (9.2%) under BP MASTER augmentation. The recall at 50% IoU for blood vessel and pancreas segmentations were 0.798 and 0.972, and a significant improvement was observed in blood vessel segmentation (10.7%) but not in the pancreas (9.3%). With BP MASTER augmentation, a significant improvement in the recall at 50% IoU was observed in blood vessel segmentation (10.7%) but not in the pancreas (9.3%). Significant precision at 50% IoU improvement was observed in pancreas segmentation (10.9%) but not in blood vessel segmentation (2.4%). Thus, Zhang et al. [39] showed the potential for station recognition and segmentation of the pancreatic and perivascular segments by AI, shortening of the learning curve in EUS education, and improving EUS quality control in the future.

Tonozuka et al. [40] developed an original CAD system using CNNs of EUS images and reported its PC detection ability, using control images from patients with CP and NP as a preliminary study to analyze whether EUS can correctly recognize pancreatic masses. The CNN in this study consisted of seven convolution layers, followed by six normalization layers to speed up the training and improve robustness, six activation layers using the ReLU, and four max-pooling layers to downsize images to half their original width and height. The output of the final layer was a score array, representing whether or not each position in the image contained a lesion. Then, to show which parts of the image the AI recognized as important, the AI was presented with color images using gradient-weighted class activation mapping (Grad-CAM) [41]. A total of 139 patients, including 76 PC, 34 CP, and 29 NP patients, were divided into two data sets; the first set included 92 patients (51 PC, 22 CP, and 19 NP patients) who were defined as the training and validation set, and the remaining 47 patients (25 PC, 12 CP, and 10 NP patients) were independently defined as the test set. After training the model with 88,320 images, ten-fold-cross validation and independent tests were conducted. For the ten-fold cross-validation data set, the sensitivity, specificity, PPV, and NPV were 90.2%, 74.9%, 80.1%, and 88.7%, respectively. For the test data set, those values were 92.4%, 84.1%, 86.8%, and 90.7%, respectively. The AUROC of the validation and test data sets were 0.924 and 0.940, respectively. Regarding misdetection, which was defined as a rate of concordance between the clinical diagnosis and the AI diagnosis of less than 90%, there were a total of 34 cases, including 13 overlooked PC cases and 21 overdiagnosed non-PC cases. In univariate analysis of the non-PC cases, they found that male sex, CP cases, mass formation, hyperechoic foci without acoustic shadow, and main pancreatic duct dilation were significant factors involved in the overdiagnosis of tumors. The multivariate logistic analysis demonstrated that only the factor of mass formation was associated with overdiagnosis of a tumor in the non-PC cases (*p* = 0.022; OR, 9.08; 95% CI, 1.37–60.00). Although previous reports have utilized EUS-CADx for tumor differentiation, this was the first report on the feasibility of EUS-CADe applications for pancreatic cancer mass detection in EUS images prior to subsequent tumor differentiation.

## 5. Discussion

The application of ML and DL to EUS imaging of the pancreas has been reported to achieve equal or better results than endoscopists, although few studies have been conducted. With further research and development, several advantages are likely to arise from this clinical application. The first is the role of optical biopsy as an alternative to tissue sampling. Currently, EUS-FNA is widely used in the diagnosis of pancreatic tumors, but there are cases in which tissue sampling cannot be performed owing to intervening vessels or other factors, or there are adverse events such as bleeding, pancreatitis, or tumor dissemination. However, once the application of CADx to EUS images is established, this will enable diagnostic estimation without such risks, which is helpful in determining treatment strategies. Secondly, the diagnostic accuracy is improved by double reading with humans; even experts can miss lesions owing to fatigue or inattention, and double reading with AI can improve their detection and diagnostic capabilities. Furthermore, if double reading is implemented in real-time for EUS, it may lead to a reduction in examination time. Thirdly, these advances can contribute to the education of inexperienced endoscopists. In general, EUS education requires a high amount of time training under experts, but AI could replace or augment that expertise and also shorten the learning curve. 

On the other hand, there are also some limitations to the application of AI to EUS imaging. This is owing to the small number of pancreatic diseases themselves, the overwhelmingly low number of EUS examinations compared to other examinations, and the need for robust techniques to produce diagnostic images. There are also challenges in dealing with rare pancreatic diseases and atypical findings. A solution to this problem requires multicenter collaboration and a database containing many images. There are already online databases with many images of chest X-rays, CT scans, and MRI fundus images that are used by AI researchers, and cooperation at the national or global level may be necessary for EUS data as well. Secondly, in AI, especially in DL, the judgment and recognition of computers are invisible, a phenomenon known as the “black box problem.” Therefore, even if a misdiagnosis occurs, there may be cases in which the basis for the decision is not understood, and depending on how it is used, the patient’s life may be entrusted to a black box, such that even the doctors and developers cannot understand the basis for their decisions. This is a fatal flaw in the context of today’s evidence-driven medicine. An expandable AI that uses Grad-CAM (Figure 5) [41] and other technologies to visualize AI decisions has been developed, and the evolution of AI requires the development of such technologies.

## 6. Conclusions

Although AI studies on EUS imaging of the pancreas are still in their infancy, AI has the potential to make great contributions to the use of EUS, which is a highly specialized imaging technique, and further research is thus desirable. In addition, advances in AI technology have simultaneously created complexity in these approaches, which requires collaboration not only with physicians but also with computer technicians and engineers from ultrasound endoscope development companies.

## Figures and Tables

**Figure 1 diagnostics-11-00018-f001:**
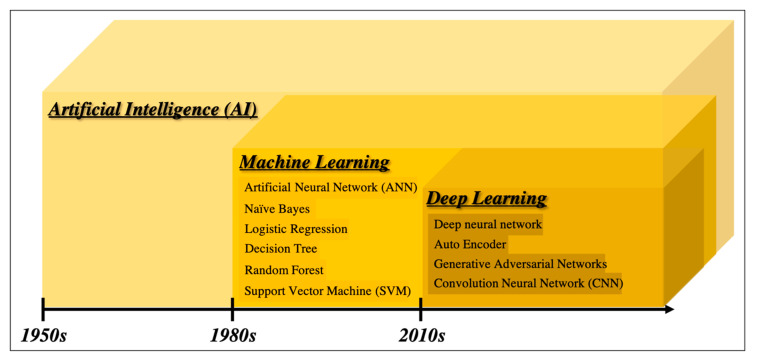
Overview of the development of artificial intelligence.

**Figure 2 diagnostics-11-00018-f002:**
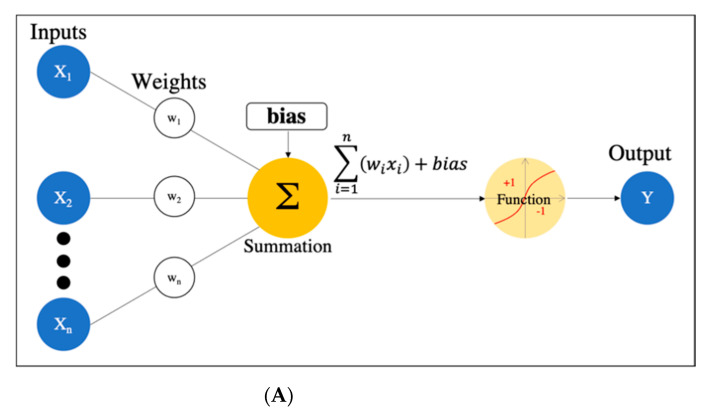
Artificial neural network. (**A**) Simple perceptron. (**B**) Neural network. (**C**) Deep neural network (DNN). Black dots: multiple hidden layers.

**Figure 3 diagnostics-11-00018-f003:**
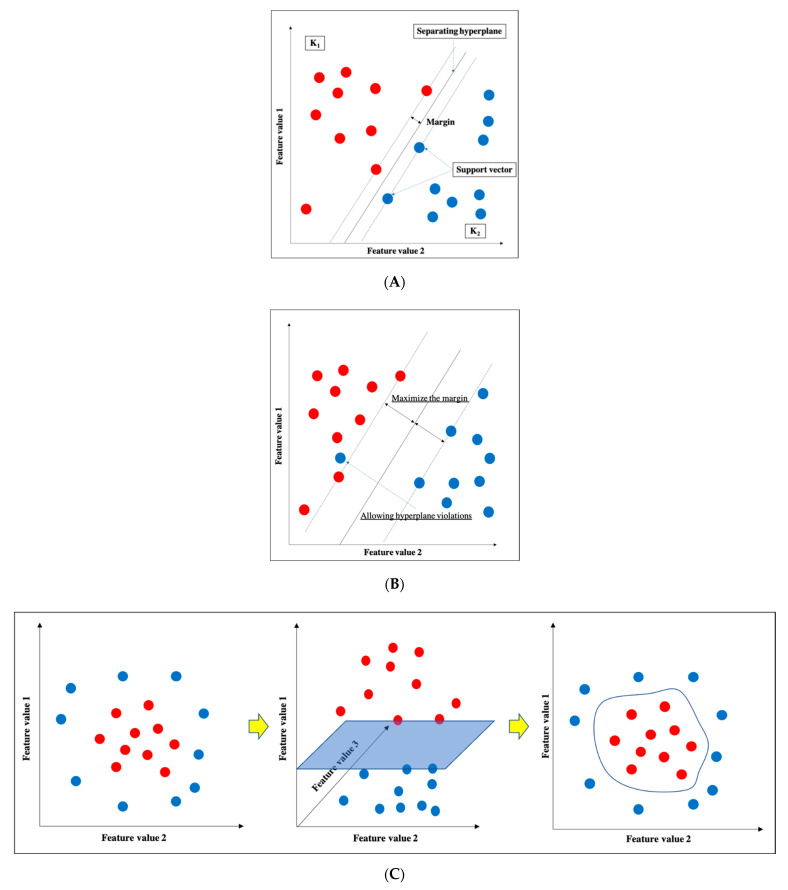
Support vector machine (SVM). (**A**) Basic structure of SVM. (**B**) Soft-margin SVM. (**C**) Kernel method.

**Figure 4 diagnostics-11-00018-f004:**
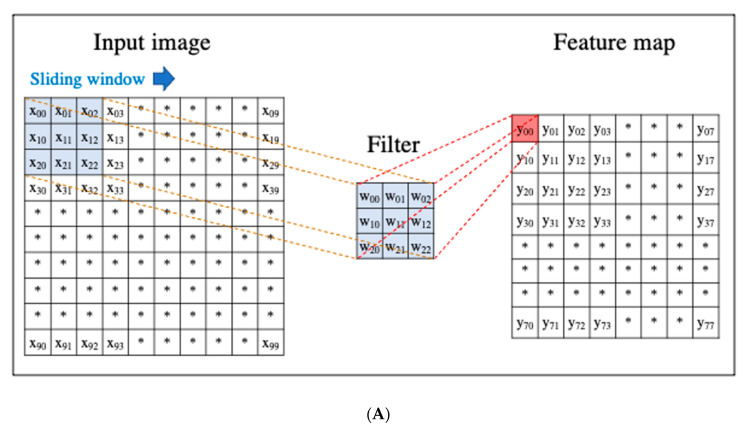
An example convolutional neural network (CNN). (**A**) Translation of an input image into a feature map in a convolution layer. (**B**) Layout of a deep convolutional neural network.

**Figure 5 diagnostics-11-00018-f005:**
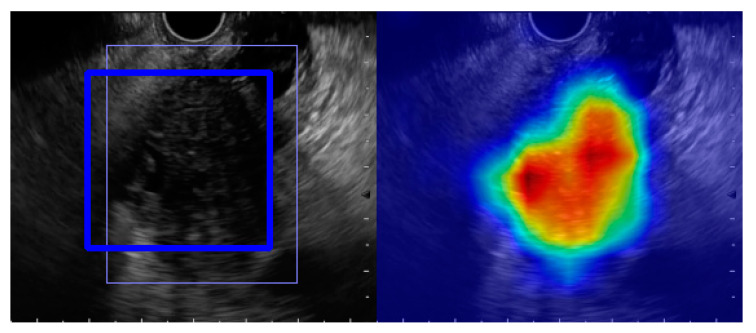
Gradient-weighted lass activation mapping (Grad-CAM). The left image is a representative original endoscopic ultrasound image. The right is a Grad-CAM image displaying the regions recognized as being important.

**Table 1 diagnostics-11-00018-t001:** Summary of reviewed published studies using artificial intelligence to analyze pancreatic endoscopic ultrasound (EUS) data.

Author	Year	Objective	Case Number	Analysis Target	Type of CAD	Algorithm of AI
Norton ID [31]	2001	Classification (PC vs. CP)	35	Grayscale pixels from B-mode image	Conventional CAD	Basic neural network
Das A [32]	2008	Classification (PC vs. CP and NP)	56	Texture features from B-mode image	Conventional CAD	ANN (multilayered perceptron)
Zhang MM [33]	2010	Classification (PC vs. CP and NP)	216	Texture features from B-mode image	Conventional CAD	SVM
Saftoiu A [34]	2012	Classification (PC vs. CP)	258	Hue histogram from EUS-elastgraphy	Conventional CAD	ANN (multilayered perceptron)
Zhu M [35]	2013	Classification (PC vs. CP)	388	Texture features from B-mode image	Conventional CAD	SVM
Saftoiu A [36]	2015	Classification (PC vs. CP)	167	Parameters of time-intensity curve from contrast-enhanced EUS	Conventional CAD	ANN
Ozkan M [37]	2016	Classification (PC vs. NP)	172	Digital features from B-mode image	Conventional CAD	ANN
Kuwahara T [38]	2019	Classification (malignant IPMN vs. benign IPMN)	50	B-mode image	Deep Learning based CAD	CNN
Zhang J [39]	2020	EUS station recognition and pancreas segmentation	480	B-mode image	Deep Learning based CAD	CNN
Tonozuka R [40]	2020	Detection of PC	139	B-mode image	Deep Learning based CAD	CNN

CAD, computer-aided diagnosis; AI, artificial intelligence; PC, pancreatic cancer; CP, chronic pancreatitis; NP, normal pancreas; ANN, artificial neural network; SVM, support vector machine; IPMN, intraductal papillary mucinous neoplasm; CNN, convolution neural network.

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
