# Peer review of "The Role of Artificial Intelligence in Endoscopic Ultrasound for Pancreatic Disorders"

_diagnostics, 2020, doi:10.3390/diagnostics11010018_

Round 1

Reviewer 1 Report

Comments

In this review, the authors provide the role of AI in EUS

  1. Please add more specificity to the title. Role of AI in EUS for pancreatic disorders. The term general review doesn’t attract an audience. 
  2. Abstract: In general well written. Authors are encouraged to provide the main reason for the review. A mention about due to a rapidly developing field and input of publication, it is critical for a sound understanding and use of potential AI in the future. Also, it is expected that AI could be incorporated for the advanced endoscopists and echoendoscopists in the near future and hence the need for this review. 
  3. Introduction: Overall well written. Line 38, use the term ASGE. Line 46, please include CRC screening (PMID: 33076511), Figure 1 needs to be improved. Though the concept is good, the font is too small. The same holds good for figure 2b, 4a, and 4b. These need to be resized. 
  4. Section 2.1, 2.2 is appropriately written. 
  5. Search strategy: Authors used on Scholar, Medline/PubMed. Can they extend the search to other engines such as OVID, Cochrane, Scopus?
  6. Section 4.1 font seems different from the rest of the article. Please make sure that no major errors have entered this section. I could not find any so far. The same holds for pages 11, 12. 
  7. Table 1: Can be extended for more columns (take the content from the text and include the table)- as some of the text paragraphs appear busy. The rest of the content is good. 
  8. Overall well-written manuscript. The authors did a good job on this.

Author Response

RESPONSE TO REVIEWER 1:

  1. Please add more specificity to the title—role of AI in EUS for pancreatic disorders. The term general review doesn’t attract an audience.

>> Thank you for your kind comments. As you suggested, we have changed the title to "Role of Artificial Intelligence in Endoscopic Ultrasound for Pancreatic Disorders” on Line 2-3.

  1. Abstract: In a general, well written. Authors are encouraged to provide the main reason for the review. A mention about due to a rapidly developing field and input of publication, it is critical for a sound understanding and use of potential AI in the future. It is also expected that AI could be incorporated for the advanced endoscopists and echoendoscopists in the near future and hence the need for this review.

>> Thank you for your comment. In the abstract, we have rewritten Lines 14-15 as follows to emphasize this review's purpose.

“This review briefly summarizes each report in order to help endoscopists better understand and utilize the potential of this rapidly developing AI,”

  1. Introduction: Overall well written. Line 38, use the term ASGE.

>> In Line 38, we replaced the term with ASGE.

Line 46, please include CRC screening (PMID: 33076511),

>> We have added a citation for PMID:33076511 on Line 51.

 Figure 1 needs to be improved. Though the concept is good, the font is too small. The same holds good for figure 2b, 4a, and 4b. These need to be resized.

>> Thank you for your helpful comments. We have reviewed and fixed the font for all images.

  1. Section 2.1, 2.2, is appropriately written.

>> Thank you.

  1. Search strategy: Authors used on Scholar, Medline/PubMed. Can they extend the search to other engines such as OVID, Cochrane, Scopus?

>> Thank you for your suggestion. We have added and checked OVID, Cochrane, and Scopus searches. Then we have written Line 250-251 as “Two authors (R.T and S.M) used the Google search engine, Google Scholar, MEDLINE via PubMed, OVID, Cochrane, and Scopus.”.

  1. Section 4.1 font seems different from the rest of the article. Please make sure that no major errors have entered this section. I could not find any so far. The same holds for pages 11, 12.

>> Thank you for your kind comments. We have fixed the font for Lines 269-605.

  1. Table 1: Can be extended for more columns (take the content from the text and include the table)- as some of the text paragraphs appear busy. The rest of the content is good.

>> Thank you for your suggestion. We have added contents in Table1.

  1. Overall well-written manuscript. The authors did a good job on this.

>> Thank you for your careful peer review.

Reviewer 2 Report

  1. This manuscript provides a general review for the artificial intelligence in diagnostic endoscopic ultrasound for the diagnosis of pancreatic disorders. It also provides a basic overview of the artificial intelligence involved. I consider the manuscript should be quite informative to the readers and suitable for publication.
  2. The Fig 2 (b): The labels were flipped over.

Author Response

RESPONSE TO REVIEWER 2:

  1. This manuscript provides a general review for the artificial intelligence in diagnostic endoscopic ultrasound for the diagnosis of pancreatic disorders. It also provides a basic overview of the artificial intelligence involved. I consider the manuscript should be quite informative to the readers and suitable for publication.

>> Thank you for your great feedback.

  1. The Fig 2 (b): The labels were flipped over.

>> Thank you for pointing this out. We have corrected it.